# Relationship between the Young’s Modulus of the Achilles Tendon and Ankle Dorsiflexion Angle at Maximum Squat Depth in Healthy Young Males

**DOI:** 10.3390/medicina59061105

**Published:** 2023-06-07

**Authors:** Hayato Miyasaka, Bungo Ebihara, Takashi Fukaya, Hirotaka Mutsuzaki

**Affiliations:** 1Department of Rehabilitation, Tsuchiura Kyodo General Hospital, 4-1-1 Otsuno, Tsuchiura 300-0028, Ibaraki, Japan; 2Department of Rehabilitation, JA Toride Medical Center, 2-1-1 Hongo, Toride 302-0022, Ibaraki, Japan; bun.hirakata@gmail.com; 3Department of Physical Therapy, Faculty of Health Sciences, Tsukuba International University, 6-8-33 Manabe, Tsuchiura 300-0051, Ibaraki, Japan; t-fukaya@tius.ac.jp; 4Center for Medical Science, Ibaraki Prefectural University of Health Sciences, Ami 300-0394, Ibaraki, Japan; mutsuzaki@ipu.ac.jp; 5Department of Orthopedic Surgery, Ibaraki Prefectural University of Health Sciences Hospital, Ami 300-0331, Ibaraki, Japan

**Keywords:** deep squatting, Achilles tendon, elastography, Young’s modulus

## Abstract

*Background and Objective*: Achilles tendon (AT) stiffness can reduce ankle dorsiflexion. However, whether AT stiffness affects the ankle dorsiflexion angle at a maximum squat depth remains unclear. Therefore, we aimed to investigate the relationship between the Young’s modulus of the AT and ankle dorsiflexion angle at the maximum squat depth in healthy young males using shear-wave elastography (SWE). *Materials and Methods*: This cross-sectional study included 31 healthy young males. AT stiffness was measured using the Young’s modulus through SWE. The ankle dorsiflexion angle at the maximum squat depth was measured as the angle between the vertical line to the floor and the line connecting the fibula head and the lateral malleolus using a goniometer. *Results*: Multiple regression analysis identified the Young’s modulus of the AT at 10° of ankle dorsiflexion (standardized partial regression coefficient [*β*] = −0.461; *p* = 0.007) and the ankle dorsiflexion angle in the flexed knee (*β* = 0.340; *p* = 0.041) as independent variables for the ankle dorsiflexion angle at maximum squat depth. *Conclusions*: The Young’s modulus of the AT may affect the ankle dorsiflexion angle at the maximum squat depth in healthy young males. Therefore, improving the Young’s modulus of the AT may help increase the ankle dorsiflexion angle at maximum squat depth.

## 1. Introduction

Deep squatting is a motion involving total body flexion that can be used to evaluate functional movement [1,2]. It is routinely performed by Asians during daily life activities and farmwork [3,4]. However, an Achilles tendon (AT) rupture can lead to an inability to perform deep squatting [5], and such an injury could negatively impact employment. Therefore, improving the deep squatting motion may help accelerate the patient’s return to work.

Ankle dorsiflexion flexibility has been shown to influence deep squatting motion [6]. Kim et al. (2015) reported that the ankle dorsiflexion range of motion (ROM) is a major factor affecting squat depth in both sexes and is critical to increasing ankle joint mobility [7]. Chang et al. (2021) also reported that ankle dorsiflexion ROM correlates with AT stiffness [8]. However, whether AT stiffness affects the ankle dorsiflexion angle at maximum squat depth remains unclear.

Methods for estimating tissue stiffness using ultrasound-based shear-wave elastography (SWE) have been proposed by previous studies [9,10]. Notably, SWE has been increasingly used to assess tendon stiffness [11,12], which is represented in terms of Young’s modulus. With the measurement of the shear-wave propagation velocity distribution in the tissues, SWE produces numerical data in kilopascals (kPa) [9]. Young’s modulus (E), which is a measure of tissue stiffness, is calculated as follows: E = 3ρc^2^; Young’s modulus, E; tissue density, ρ; and shear wave velocity, c [13]. An elevated Young’s modulus signifies the stiffness of the tissue. SWE can measure the Young’s modulus of an AT with high reliability [11,14]; therefore, assessing the Young’s modulus of the AT may be a useful method for predicting the ankle dorsiflexion angle at maximum squat depth. Additionally, SWE is expected to be useful in the evaluation of rehabilitation programs, provide an early diagnosis, and detect injury risk in the assessment of tendon pathology [15]. An AT rupture occurs most frequently in young males [16]. Therefore, evaluating the Young’s modulus of the AT in young males may help prevent AT rupture.

Therefore, this study aimed to clarify the relationship between the Young’s modulus of the AT and the ankle dorsiflexion angle at maximum squat depth in healthy young males. We hypothesized that the Young’s modulus of the AT affects the ankle dorsiflexion angle at the maximum squat depth. Furthermore, if this hypothesis is proven, improving the Young’s modulus of the AT may help increase the ankle dorsiflexion angle at the maximum squat depth.

## 2. Materials and Methods

### 2.1. Participants

Overall, 31 healthy young males participated in this study. The participants’ characteristics are summarized in Table 1. The dominant leg was assessed by asking the participants which leg they would use to kick a ball [17]. The right foot of all participants was measured. Participants were included if they had a maximum ankle dorsiflexion ROM of at least 10°; had no fever, joint pain, or muscle pain; and could understand and sign the consent form. In contrast, participants were excluded if they had a history of neuromuscular disease or musculoskeletal injury to the lower limbs. The same physical therapist, with 5 years of experience on musculoskeletal ultrasound examinations, assessed each participant for tendon health through palpation and B-mode ultrasound. None of the participants exhibited tendinopathy-related swelling or hypoechoic areas in the tendon.

This study was approved by the ethics committee of the institute, and the study protocol was conducted in accordance with the Declaration of Helsinki.

### 2.2. Measurement of Ankle Dorsiflexion ROM

The patients were placed supine, and ankle dorsiflexion ROM was measured in the extended- and flexed-knee positions using a goniometer with a minimum value of 1°. During measurement, the fulcrum of the goniometer was centered over the lateral malleolus, the stationary arm was aligned with the long axis of the fibula, and the movement arm was parallel to the fifth metatarsal.

### 2.3. Measurement of the Ankle Dorsiflexion Angle at the Maximum Squat Depth

The participants were instructed to bodyweight squat to the maximal depth position they could maintain for 3 s to measure the ankle dorsiflexion angle at maximum squat depth. They were also instructed to stand with their feet shoulder-width apart with their arms extended forward, parallel to the floor, while maintaining a straight line of vision. Participants were instructed to maintain heel contact during the task. Ankle dorsiflexion was measured using a goniometer as the angle between the vertical line to the floor and the line connecting the fibula head and lateral malleolus (Figure 1), with a minimum value of 1°.

### 2.4. Measurement of Ankle Plantar/Dorsiflexion Strength

The ankle joints’ muscles were assessed for strength using a Biodex 3 dynamometer (Biodex Medical Systems, Shirley, NY, USA). The participants were in a sitting position, with 30° of knee flexion. The lower trunk, thigh, and ankle muscles were stabilized using straps. Bilateral isokinetic (concentric/concentric) ankle plantar/dorsiflexion measurements were performed, and two sets of five maximal dynamic repetitions were conducted at an angular velocity of 60°/s, with a 30-s interval between these sets. Each participant was positioned with their feet parallel to the floor to avoid hamstring action during the test. Finally, the peak torque was measured at a minimum of 1 Nm, and peak torque/body weight was calculated.

### 2.5. Measurement of the Young’s Modulus of the AT

Ultrasound examinations were performed using a 2–10 MHz linear transducer (Supersonic Imaging, Aix-en-Provence, France). The same physical therapist, with 5 years of experience in musculoskeletal ultrasound examinations, assessed the Young’s modulus using the SWE Opt penetration mode. The Young’s modulus range for the AT was 0–800 kPa. The participants were measured in the kneeling position with 90° of knee flexion and the upper body supported by a table (Figure 2). We instructed the participants to remain relaxed during the measurements. Ultrasound images were recorded along the longitudinal axis of the tendon at ankle dorsiflexion angles of −10°, 0°, and 10°. The ankle joint was securely attached to the tilt table footplate. AT fibers and the transducer were made parallel and the AT region was defined as 3 cm above the calcaneal tuberosity [18]. Furthermore, the skin surface was marked with a black pen, and the probe was kept stationary for 10 s with a 3-mm-diameter region of interest (ROI) circle [19]. The ROI was set near the center of the AT (Figure 3). Therefore, to reduce the effect of pressure, a generous amount of gel was applied, and the probe was carefully positioned over the skin. However, measurements of each ankle position were taken at −10°, 0°, and 10° of dorsiflexion in the specified order to lessen the impact of stretching.

Fifteen minutes after the initial measurement, the measurements were repeated to assess reliability [12]. The average Young’s modulus of the first and second measurements was used to calculate the intraclass correlation coefficient (ICC). The ICC (1.1) values of the Young’s modulus at −10°, 0°, and 10° of ankle dorsiflexion were 0.968, 0.959, and 0.958, respectively.

### 2.6. Statistical Analysis

A power analysis of the multiple regression, with an effect size *f*^2^ of 0.35, an error probability of 0.05, a power of 0.8, and two predictors, was performed using the ankle dorsiflexion angle at maximum squat depth as the dependent variable. Using G*Power 3.1 [20], a power analysis was conducted, and 31 participants were included in the study. The distribution of the measured values was assessed using the Shapiro–Wilk test. Means and standard deviation were used to describe normally distributed data; otherwise, the medians and interquartile ranges were computed.

One-way repeated measures, analysis of variance, and Bonferroni post hoc test were used to demonstrate the effects of ankle position. Pearson’s product–moment correlation coefficients were used to analyze the correlations between relationships. The ankle dorsiflexion angle at the maximum squat depth was used as the dependent variable in the stepwise multiple regression analysis.

Statistical significance was set at *p* < 0.05. All statistical analyses were performed using SPSS^®^ statistics version 24.0 (IBM Corp., Armonk, NY, USA).

## 3. Results

### 3.1. Participants’ Measured Values

The participants’ measured values are summarized in Table 2. The mean ankle dorsiflexion ROM in the extended knee, the flexed knee, and at the maximum squat depth was 16.5 ± 3.3°, 22.8 ± 3.6°, and 38.8 ± 7.0°, respectively.

### 3.2. Young’s Modulus of the AT at −10°, 0°, and 10° of Ankle Dorsiflexion

The differences in the Young’s modulus between the ankle positions are summarized in Figure 4. The mean Young’s modulus of the AT at −10°, 0°, and 10° of ankle dorsiflexion was 323.9 ± 48.7°, 400.1 ± 45.6°, and 501.7 ± 32.6°, respectively. One-way repeated measures analysis of variance showed the significant main effects of the ankle position. The Young’s modulus varied significantly among all angles (*p* < 0.001), according to post hoc analysis, and the values of the Young’s modulus increased as the ankle dorsiflexed (Figure 4).

### 3.3. Correlation Coefficients

The correlations between the ankle dorsiflexion angle at the maximum squat depth and other measurement values are shown in Table 3. The ankle dorsiflexion angle at maximum squat depth correlated with the Young’s modulus of the AT at 10° of ankle dorsiflexion (*r* = −0.640; *p* < 0.001), ankle dorsiflexion ROM in the extended knee (*r* = 0.441; *p* = 0.013), and ankle dorsiflexion ROM in the flexed knee (*r* = 0.583; *p* < 0.001). The other measurement parameters were not significantly correlated (*p* > 0.05).

### 3.4. Multiple Regression Analysis

The results of the multiple regression analysis are presented in Table 4. The Young’s modulus of the AT at 10° (*p* = 0.007) and the ankle dorsiflexion angle in the flexed knee (*p* = 0.041) were selected as independent variables. The multiple correlation coefficient, coefficient of determination (*R*^2^), and adjusted coefficient of determination (adjusted *R*^2^) were 0.702, 0.493, and 0.457, respectively. The Durbin–Watson ratio was 2.169, and the residuals were normally distributed (*p* = 0.366). The regression formula is as follows:

Ankle dorsiflexion angle at maximum squat depth (°) = 66.844–0.089 × Young’s modulus of the AT at 10° of ankle dorsiflexion (kPa) + 0.596 × ankle dorsiflexion ROM in the flexed knee (°).

## 4. Discussion

We investigated the relationship between the Young’s modulus of the AT and the ankle dorsiflexion angle at the maximum squat depth in healthy young males using SWE. The results showed that the ankle dorsiflexion angle at the maximum squat depth may reduce as the Young’s modulus of the AT increases in healthy young males.

It has been suggested that deep squatting requires a large ankle dorsiflexion ROM. However, ankle dorsiflexion ROM may be influenced by AT stiffness [21], capsular tightness, soft tissue restrictions of the foot, and posterior lower leg musculature [22]. Therefore, if the Young’s modulus of the AT of ankle dorsiflexion at 10° increases, it may lead to limited ankle dorsiflexion movement, which would make it more difficult to bend the ankle to the observed ankle dorsiflexion angle of 23.4°–25.9° at the maximum squatting depth [23].

Moreover, the Young’s modulus of the AT increased as the ankle dorsiflexed. The three-headed triceps surae, which comprises the gastrocnemius and soleus, works to plantar flex the ankle joint through its associated tendon, which is the AT [24]. The gastrocnemius and soleus are elongated during ankle dorsiflexion [25]; therefore, a passive tension is generated in the AT with ankle dorsiflexion, which can increase the Young’s modulus of the AT.

Clinically, assessing the Young’s modulus of the AT using SWE may help identify areas of AT stiffness. The AT is the biggest tendon in the human body and is one of the most frequently injured tendons [26]. Tavares et al. (2014) reported that fibrosis in the tendon and surrounding area causes AT to stiffen and lose its elasticity after healing [27]. Consequently, ankle dorsiflexion is limited after an AT rupture. Regarding acute AT ruptures, early rehabilitation is encouraged to promote functional improvement [28,29]. Therefore, rehabilitation after AT rupture may improve the Young’s modulus of stiff areas of the AT by increasing the flexibility of the three-headed triceps surae to increase the ankle dorsiflexion angle at maximum squat depth.

This study had some limitations. First, this study only included healthy young males; therefore, our findings cannot be generalized to the entire population. Therefore, corresponding studies with participants of different ages and sexes should be conducted in the future. Second, electromyography was not used during SWE measurements. Consequently, monitoring muscle activity is necessary to ensure no muscle contraction. Third, Young’s modulus only considers the AT. Therefore, the Young’s modulus of muscles, such as the triceps surae, should also be examined to discover a factor that confirms the relationship between ankle dorsiflexion angle at maximum squat depth and the Young’s modulus of the AT is specifically related to the tendon. Finally, inter-rater reliability was not assessed. The measurement of the Young’s modulus depends on the skill of the examiner [30]. Therefore, whether the measured Young’s modulus of the AT is comparable between the different examiners is unclear. Further studies are warranted to investigate the Young’s modulus of AT in individuals with AT disorders.

## 5. Conclusions

The Young’s modulus of the AT may affect the ankle dorsiflexion angle at the maximum squat depth in healthy young males. Clinically, improving the Young’s modulus of the AT may help increase the ankle dorsiflexion angle at maximum squat depth.

## Figures and Tables

**Figure 1 medicina-59-01105-f001:**
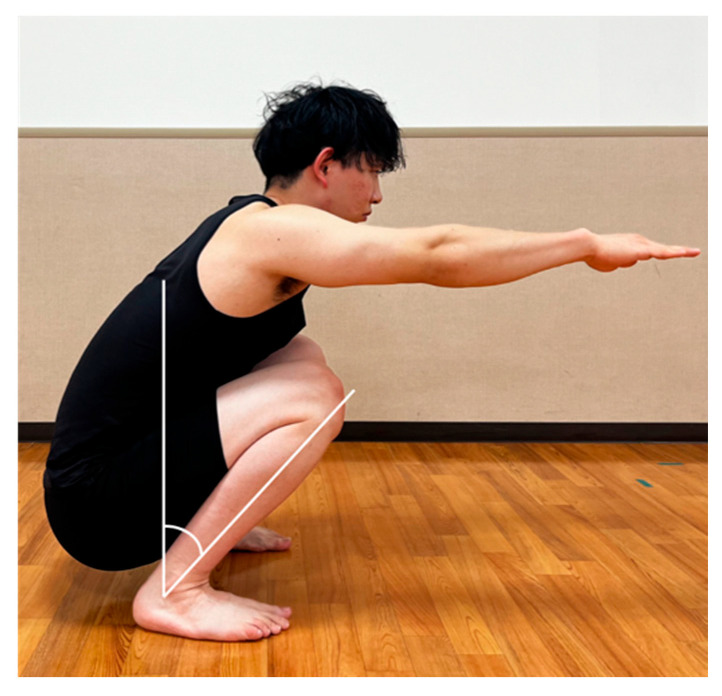
Measurement of ankle dorsiflexion angle at the maximum squat depth. Ankle dorsiflexion was measured as the angle between the vertical line to the floor and the line connecting the fibula head and lateral malleolus.

**Figure 2 medicina-59-01105-f002:**
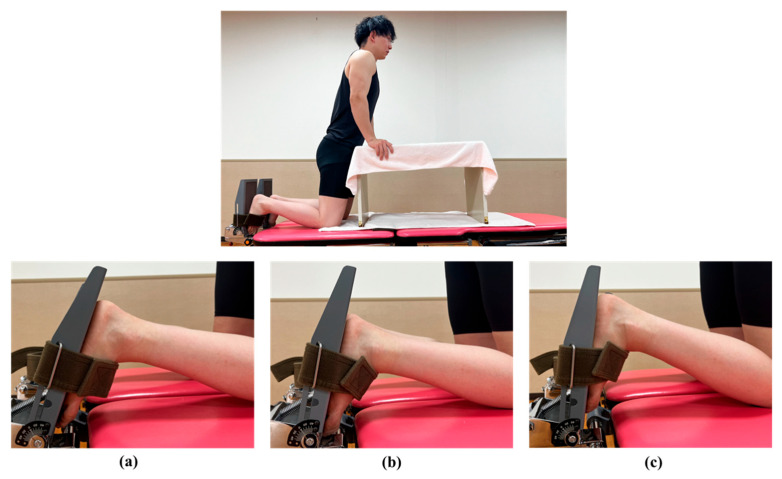
Measurement position of the Young’s modulus of the Achilles tendon during passive ankle dorsiflexion at (**a**) −10°, (**b**) 0°, and (**c**) 10°. The participants were measured while kneeling, with 90° of knee flexion and the upper body supported by a table. The ankle joint was attached to the tilt table footplate.

**Figure 3 medicina-59-01105-f003:**
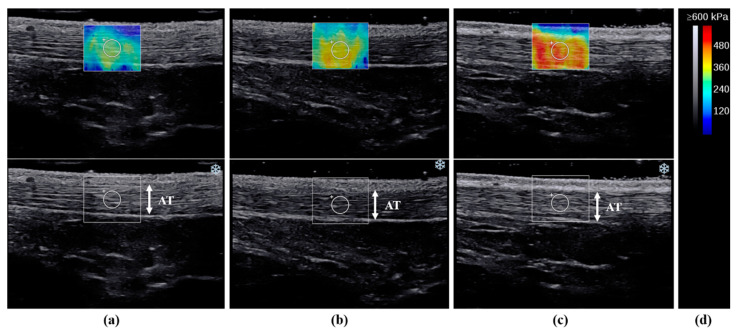
Representative images of the elasticity maps of the Achilles tendon (AT) during passive ankle dorsiflexion at (**a**) −10°, (**b**) 0°, and (**c**) 10°. (**d**) Grey and color scale. Images represent the changes in color according to stiffness and region of interest position.

**Figure 4 medicina-59-01105-f004:**
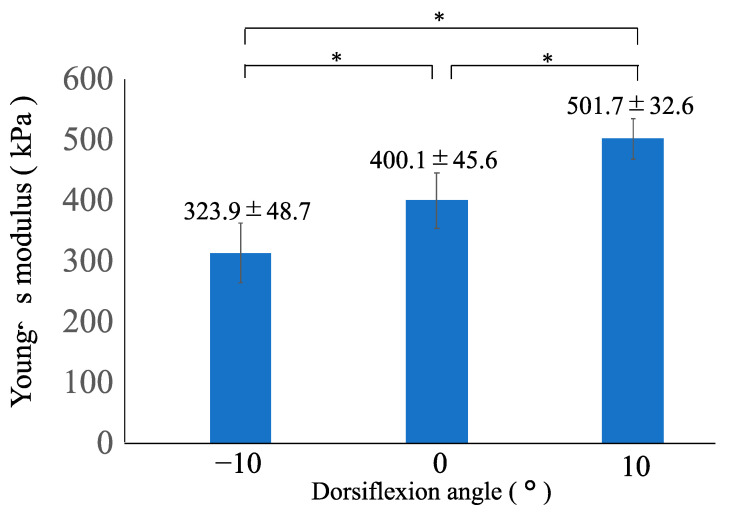
Comparison of the Young’s modulus of the Achilles tendon at −10°, 0°, and 10° of ankle dorsiflexion. * One-way repeated measures analysis of variance and Bonferroni post hoc test detected significant differences among all angles (*p* < 0.001). The Young’s modulus is represented as the mean ± standard deviation.

**Table 1 medicina-59-01105-t001:** Participants’ physical characteristics (*n* = 31).

Age (years)	27.0 (25.0–34.0)
Height (m)	1.73 (1.70–1.75)
Weight (kg)	64.3 (59.9–68.0)
Body mass index (kg/m^2^)	21.5 (20.7–22.6)
Dominant leg (right/left)	28/3

Values are presented as medians (interquartile ranges) or *n*/*n*.

**Table 2 medicina-59-01105-t002:** Participants’ measured values (*n* = 31).

Ankle dorsiflexion range of motion (°)	
Extended knee	16.5 ± 3.3
Flexed knee	22.8 ± 3.6
Maximum squat depth	38.8 ± 7.0
Ankle strength (Nm/kg)	
Plantar flexion	0.5 ± 0.1
Dorsiflexion	1.0 ± 0.3

Values are presented as means ± standard deviation.

**Table 3 medicina-59-01105-t003:** Pearson’s correlation coefficients between the ankle dorsiflexion angle at maximum squat depth and other measurement values.

Measurements	Ankle Dorsiflexion Angle at Maximum Squat Depth	
	*r*	*p* Value
Ankle dorsiflexion range of motion		
Extended knee	0.441	0.013 **
Flexed knee	0.583	<0.001 *
Young’s modulus of the Achilles tendon	−0.640	<0.001 *
Ankle strength		
Plantar flexion	0.181	0.329
Dorsiflexion	−0.006	0.973

* Correlation was significant at *p* = 0.01. ** Correlation was significant at *p* = 0.05.

**Table 4 medicina-59-01105-t004:** Multiple regression analysis results.

Variables	*B*	95% CI of *B*	*p*-Value	*β*	VIF
Constant	66.844	26.868–106.820	0.002		
Young’s modulus of the Achilles tendon (kPa)	−0.089	−0.152 to −0.026	0.007	−0.461	1.386
Ankle dorsiflexion range of motion in the flexed knee (°)	0.596	0.027–1.165	0.041	0.340	1.386

*B:* partial regression coefficient; *CI*: confidence interval; *β:* standardized partial regression coefficient; *VIF:* variance inflation factor.

## Data Availability

The data that support the findings of this study are available on request from the corresponding author. The data are not publicly available, owing to restrictions on their containing information that could compromise the privacy of research participants.

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
