# Peer review of "Relationship between the Young’s Modulus of the Achilles Tendon and Ankle Dorsiflexion Angle at Maximum Squat Depth in Healthy Young Males"

_medicina, 2023, doi:10.3390/medicina59061105_

Round 1

Reviewer 1 Report

Dear Authors,

your manuscript entitled "Relationship between the Young’s modulus of the Achilles tendon and ankle dorsiflexion angle at maximum squat depth in healthy young males" is well-written, well-presented and the methodology is adequate.

I would suggest a few things to improve the quality of the article:

1) please explain what the Young's modulus is in the Introduction section;

2) In the introduction section, you stated that "Such an injury could negatively impact employment. Improving the deep squatting motion may help accelerate the patient’s return to work".

I would suggest to explain how your outcomes can be translated in the clinical practice. How can your results be useful for health practitioners? 

For example, may the assessment of the Young's modulus in active subjects prevent an Achilles tendon rupture?

3) Please cite this article when talking about AT rehabilitation after a rupture:

Tarantino D, Palermi S, Sirico F, Corrado B. Achilles Tendon Rupture: Mechanisms of Injury, Principles of Rehabilitation and Return to Play. Journal of Functional Morphology and Kinesiology [Internet] 2020;5(4):95. Available from: http://dx.doi.org/10.3390/jfmk5040095

Author Response

We thank the reviewers for their constructive critique to improve the manuscript. Please see the attachment.

Reviewer 2 Report

The purpose of this study was to investigate the relationship between the Young's modulus of the Achilles tendon (AT) and ankle dorsiflexion angle at the maximum squat depth in healthy young males using shear-wave elastography (SWE).

Two specific points were considered in this study:

1. The study aimed to evaluate the elastic characteristics of the Achilles tendon, considering its well-known viscoelastic nature within the human body. The primary focus was on designing a procedure to determine the Young's modulus as a measure of its elastic properties. How to design the procedure before proceeding with the study?

2. To account for the potential influence of the anterior tibialis muscle on the ankle dorsiflexion angle during deep squat conditions, efforts were made to eliminate its impact. How to ensure that the relationship observed between the Young's modulus of the Achilles tendon and ankle dorsiflexion angle specifically pertained to the tendon itself?

To gain a better understanding of the relationship between the Young's modulus of the Achilles tendon and the ankle dorsiflexion angle at the maximum squat depth in healthy young males, providing additional details about the methodology and results of the study would be helpful.

Author Response

(The authors gave the same response as above.)
